# Role of the volume conductor on simulations of local field potential recordings from deep brain stimulation electrodes

**M. Sohail Noor**[1], **Bryan Howell**[1], **Cameron C. McIntyre**[1,2]*

**1** Department of Biomedical Engineering, Duke University, Durham, NC, United States of America,
**2** Department of Neurosurgery, Duke University, Durham, NC, United States of America

* cameron.mcintyre@duke.edu

## Abstract

### Objective

Local field potential (LFP) recordings from deep brain stimulation (DBS) electrodes are commonly used in research analyses, and are beginning to be used in clinical practice. Computational models of DBS LFPs provide tools for investigating the biophysics and neural synchronization that underlie LFP signals. However, technical standards for DBS LFP model parameterization remain to be established. Therefore, the goal of this study was to evaluate the role of the volume conductor (VC) model complexity on simulated LFP signals in the subthalamic nucleus (STN).

### Approach

We created a detailed human head VC model that explicitly represented the inhomogeneity and anisotropy associated with 12 different tissue structures. This VC model represented our "gold standard" for technical detail and electrical realism. We then incrementally decreased the complexity of the VC model and quantified the impact on the simulated LFP recordings. Identical STN neural source activity was used when comparing the different VC model variants.

### Results

Ignoring tissue anisotropy reduced the simulated LFP amplitude by ~12%, while eliminating soft tissue heterogeneity had a negligible effect on the recordings. Simplification of the VC model to consist of a single homogenous isotropic tissue medium with a conductivity of 0.215 S/m contributed an additional ~3% to the error.

### Significance

Highly detailed VC models do generate different results than simplified VC models. However, with errors in the range of ~15%, the use of a well-parameterized simple VC model is likely to be acceptable in most contexts for DBS LFP modeling.

**Data Availability Statement:** The VC models and raw data underlying the results presented in the study are available at https://github.com/

msohailnoor/LFPs-simulated-using-different-volume-conductor-models.

**Funding:** This work was supported by a grant from the National Institutes of Health (R01 NS119520). The funders had no role in study design, data collection and analysis, decision to publish, or preparation of the manuscript.

**Competing interests:** CCM is a paid consultant for Boston Scientific Neuromodulation, receives royalties from Hologram Consultants, Neuros Medical, Qr8 Health, and is a shareholder in the following companies: Hologram Consultants, Surgical Information Sciences, BrainDynamics, CereGate, Cardionomic, Enspire DBS. This does not alter our adherence to PLOS ONE policies on sharing data and materials.

## Introduction

Local field potential (LFP) recordings from deep brain stimulation (DBS) electrodes are commonly used in clinical research studies [1]. From a scientific perspective, LFP recordings enable identification of electrophysiological biomarkers that can be used to better understand the pathophysiology of a brain disorder [2]. From an engineering perspective, an LFP-based biomarker can be used as a control signal for the modulation of stimulation in an adaptive DBS system [3]. As such, the technical capabilities of clinical DBS devices are evolving to provide opportunities to use LFP signals in both diagnostic and therapeutic applications [4]. However, the biophysical details underlying LFP recordings from DBS electrodes are only beginning to emerge [5].

An improved understanding of the neural activity patterns and electrical conduction physics that give rise to an LFP signal should enhance opportunities to dissect the pathophysiology, as well as improve the engineering design of adaptive DBS systems. Therefore, our group has been working to create detailed biophysical models of LFP recordings with DBS electrodes [5–7]. However, our previous modeling efforts ignored the complexities of inhomogeneity and anisotropy in the brain tissue medium on LFP signal conduction to the DBS recording contacts.

The electrical conductivity of the brain can be estimated with a volume conductor (VC) model. VC models can be highly simplified (e.g. an infinite homogeneous and isotropic medium) or especially complex (e.g. explicit representation of many different tissue types and their individual electrical characteristics). Complex VC models are typically constructed as finite element models, and the VC model is a key component of electrical simulations of DBS. When attempting to quantify the neural response to DBS electric fields, the specific parameterization of the VC model can dramatically influence the simulation results [8]. However, it is unclear if the details of the VC model are similarly important when modeling LFP recordings from DBS electrodes. Previous analyses of electroencephalography (EEG) recording models have concluded that explicit representation of the cerebral spinal fluid (CSF) spaces [9], as well as uncertainties in the conductivity of the skin and skull [10], have significant influences on the simulated EEG signals. Therefore, the goal of this project was to examine the role of the VC model on simulated LFP signals, where we specifically analyzed beta-band (12–30 Hz) LFP activity recorded from subthalamic DBS electrodes.

Subthalamic beta-band activity is the most intensely studied LFP biomarker for parkinsonian symptoms, and its use as a control signal in adaptive DBS systems is beginning to transition out of the research environment and into clinical settings [11]. In addition, subthalamic DBS delivered at electrode locations that also exhibit a high degree of beta-band LFP activity are typically associated with good therapeutic outcomes [12]. Therefore, one application for patient-specific DBS LFP simulations is to use the model system to predict the spatial localization of a volume of synchronous neurons that give rise to the beta-band activity [6]. That process relies on inverse modeling to localize the synchronized neural population and enable definition of a patient-specific electrophysiology-based target volume for stimulation. That target volume can then be used to optimize selection of the DBS electrode contact(s) for stimulation, as well as their stimulation parameter settings [13]. Along that line, the results of this study provide insight on the level of detail that is appropriate for the VC model in patient-specific DBS LFP simulations. We found that tissue anisotropy had a relatively small effect, and soft tissue heterogeneity had almost no effect, on the simulated LFPs. These results suggest that simplified VC models represent a reasonable option for use in subthalamic LFP analyses.

## Methods

The overall LFP model system used in this study comprised of two main components: 1) the volume conductor (VC), which was a finite element model of the human head including the DBS lead, and 2) the neural sources, which were multi-compartment cable models that simulated transmembrane current sources associated with the electrical activity of individual neurons. The neural source models were integrated with the VC model with a reciprocity-based solution, which enabled simulation of the electrical voltage recorded at the DBS electrode contacts [5]. The goal of this study was to evaluate the role of the VC model on the simulated LFP recordings. We elected to use a highly detailed VC model as our "gold standard" for technical detail and electrical realism. We then incrementally decreased the complexity of that VC model and quantified its impact on the simulated LFP recordings. The simulations used identical neural source activity when comparing the different VC model variants.

### Volume conductor model

The base VC model was designed to represent a detailed human head with explicit representation of the inhomogeneity and anisotropy associated with different tissue structures (Fig 1; Table 1). The foundation for the base VC model was the multimodal image-based detailed anatomical (MIDA) representation of the human head [14]. The original MIDA head model consisted of 116 different anatomical structures. However, we consolidated those components into 12 core structures for DBS modeling, which we called MIDA12 [8].

The MIDA12 model provides a platform to represent the inhomogeneity of the human head. We then employed two versions of MIDA12 in our analyses: isotropic and anisotropic. MIDA12 isotropic used unique isotropic conductivities for each structure in the head model. MIDA12 anisotropic incorporated anisotropic conductivity tensors into the brain tissue of the head model, while all other structures were modeled with isotropic conductivities [8].

MIDA12 isotropic was also simplified into less heterogeneous models we called MIDA4, MIDA2, and MIDA1 (Fig 1; Table 1). MIDA4 comprised of four tissue types (CSF, grey matter, white matter, and all other tissues combined), MIDA2 consisted of two tissue types (CSF and all other tissues combined), and MIDA1 had a single homogeneous and isotropic conductive medium. When creating MIDA4, MIDA2, and MIDA1, we combined two or more tissues by assigning the consolidated volume a weighted-average conductivity from the various structures combined (Table 1). For example, in MIDA2, grey matter, white matter and CSF were combined into a composite volume whose conductivity was calculated as follows:

$$\frac{5.3 \times 1.5 + 14.85 \times 0.23 + 12.83 \times 0.14}{5.3 + 14.85 + 12.83} = 0.4 \text{ S/m}$$

Similarly, in MIDA1, all 12 tissues were combined, and their weighted-average conductivity was computed to be 0.215 S/m.

Each VC model of the human head and DBS electrode was represented as a finite element model that was created using COMSOL v5.4. The DBS electrode (Medtronic 3389) shaft was surrounded by a 0.1 mm interface layer, mimicking tissue encapsulation around the implanted electrode [5]. The MIDA12, MIDA4, MIDA2, and MIDA1 representations of the human head each had a different electric load at the DBS electrode contacts. Therefore, we adjusted the conductivity of a thin tissue interface layer (i.e. encapsulation layer) to normalize the electric load across all of the VC models. The total electric load at contact 1 was set to 1084 Ω for each VC model. The conductivity of the interface layer used in each VC model variant is provided in Table 2.

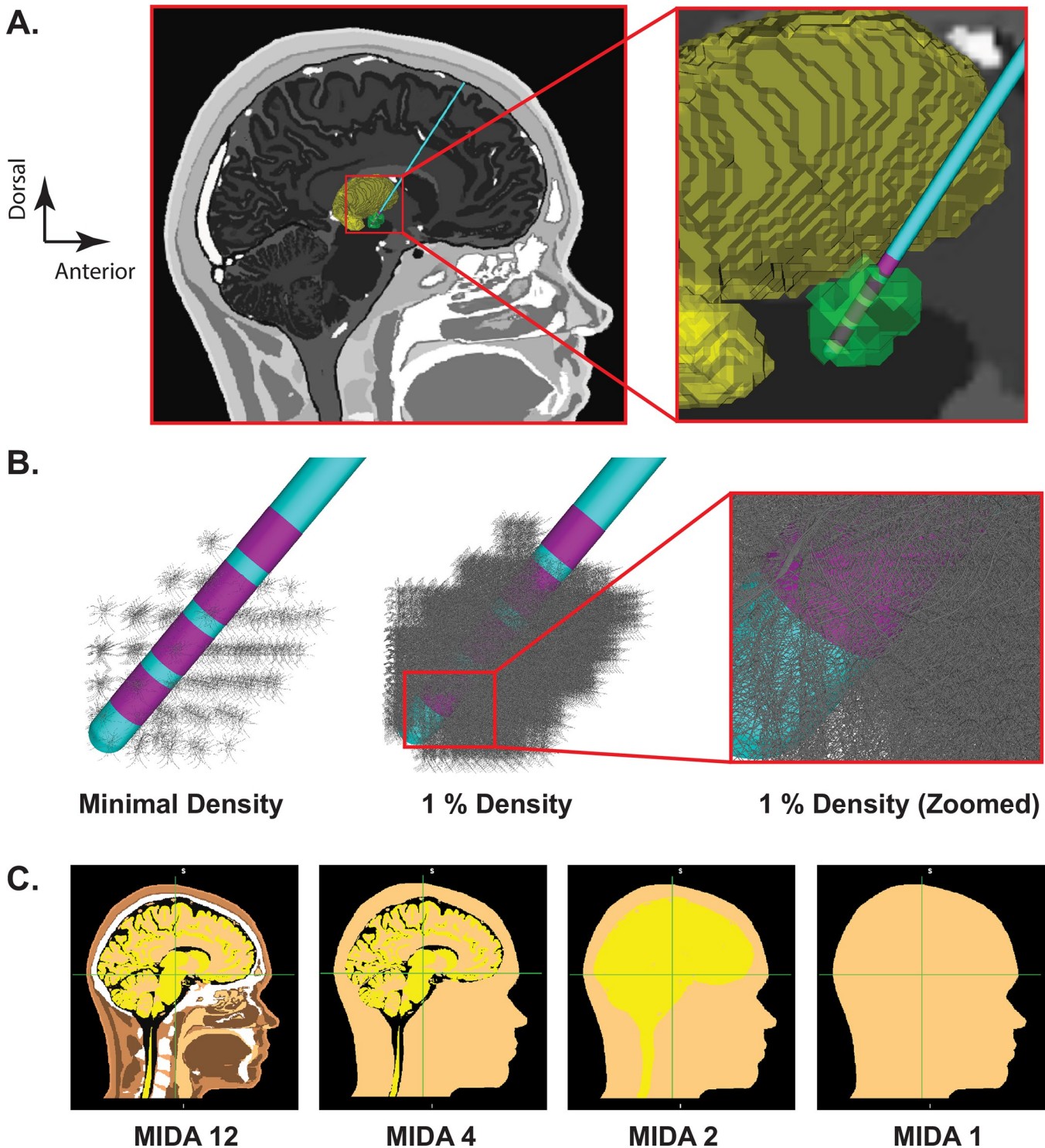

**Fig 1. Anatomical model.** A) Far left panel shows a sagittal view of the MIDA head model, deep brain stimulation (DBS) electrode, and 3D anatomical volumes representing the subthalamic nucleus (STN—green) and thalamus (yellow). B) STN neuron models surrounding the DBS lead (purple electrode contacts). Each grey STN neuron model is displayed with its full 3D geometry of the soma-dendritic architecture. Far left panel, one model neuron is shown in each voxel of the STN volume. Middle panel, 1% density of the STN neurons models, and right-most panel displays a zoomed-in view near contact 0. C) Different VC models: complexity reduces from left to right.

**Table 1. Tissue composition and conductivities for the VC models.**

| Tissues in MIDA12 | % tissue in MIDA12 | σ in MIDA12 (S/m) | σ in MIDA4 (S/m) | σ in MIDA2 (S/m) | σ in MIDA1 (S/m) |
|---|---|---|---|---|---|
| CSF | 5.3 | 1.5 | 1.5 | 0.4 | 0.215 |
| Grey matter | 14.85 | 0.23 | 0.23 | 0.4 | 0.215 |
| White matter | 12.83 | 0.14 | 0.14 | 0.4 | 0.215 |
| Dura | 2.02 | 0.03 | 0.124 | 0.124 | 0.215 |
| Muscle | 19.8 | 0.32 | 0.124 | 0.124 | 0.215 |
| Tendon | 1.1 | 0.38 | 0.124 | 0.124 | 0.215 |
| Bone | 13.93 | 0.02 | 0.124 | 0.124 | 0.215 |
| Fat | 20.94 | 0.0224 | 0.124 | 0.124 | 0.215 |
| Skin | 5.97 | 0.0002 | 0.124 | 0.124 | 0.215 |
| Disk | 0.064 | 0.65 | 0.124 | 0.124 | 0.215 |
| Blood | 1.01 | 0.7 | 0.124 | 0.124 | 0.215 |
| Air | 2.12 | 1e-12 | 0.124 | 0.124 | 0.215 |

## Subthalamic nucleus model

The location of the subthalamic nucleus (STN) was estimated in the MIDA head model by transformation of a probabilistic atlas of the STN [15], originally defined in MNI152 space, into MIDA space. This was accomplished using a 12-parameter affine transformation. A STN volume was then defined by thresholding a unique structure from the probabilistic atlas into a volume of ~150 mm$^3$ [16] (Fig 1). The DBS lead was positioned such that contact 1 was located in the center of the STN on the right side of the brain (Fig 1).

The estimated STN volume provided spatial boundaries to position STN neuron source models within the VC models. The STN volume was populated with 221,601 multi-compartment cable models of STN neurons [17]. The cells were oriented parallel to the long axis of the nucleus [18, 19]. The density and distribution of these neuron models in the STN volume were defined to be consistent with human histological measurements [6, 20] (details provided in S1 Table in S1 File).

The geometry of the individual STN neuron models were based on anatomical reconstructions of macaque STN neurons [21]. The electrical properties of the neuron models were parameterized to mimic experimentally defined transmembrane currents and action potential firing characteristics of STN neurons [17, 22]. The collection of ion channels and their conductances are specified in S2 Table in S1 File. Each STN neuron model also received 290 different synaptic input currents distributed over its structure [5]. These synaptic currents were intended to generically represent the thousands of synapses that contribute to the neural activity of an individual STN neuron. The somatic and proximal dendritic compartments of each neuron model received inhibitory synaptic input while distal compartments received excitatory synaptic input, with the inhibitory currents being slightly delayed [23]. While not

**Table 2. Conductivity of the interface layer.**

| | Conductivity (S/m) |
|---|---|
| MIDA 12 Anisotropic | 0.032 |
| MIDA 12 Isotropic | 0.030 |
| MIDA 4 | 0.030 |
| MIDA 2 | 0.022 |
| MIDA 1 | 0.030 |

explicitly modeled as such, the excitatory and inhibitory synaptic inputs could be loosely considered to represent hyperdirect and pallidal input streams to the STN neurons.

Each of the STN neuron models received unique time varying synaptic inputs. The STN neurons were designated to receive either a synchronous beta pattern of synaptic inputs, or an asynchronous pattern of synaptic inputs [6]. For the beta synchronous population of neurons, synaptic inputs were generated every 50 ms (i.e. 20 Hz) with temporal jitter that was randomly chosen from a normal distribution with a standard deviation of 6.25 ms. For the asynchronous population, each neuron received synaptic inputs at a rate randomly taken from an exponential distribution with a mean and standard deviation of 50 ms (i.e. 20 Hz). The Python functions employed to generate temporal jitter for synchronous and asynchronous neurons are provided in the supporting information. Neurons in the beta synchronous pool exhibited highly correlated activity, while the neurons in the asynchronous pool exhibited uncorrelated activity [6]. As such, the LFP signals primarily result from the synchronous neurons, while the asynchronous neurons primarily contribute to noise. The specific locations in the STN volume of the synchronous and asynchronous neurons is represented in the figures by green and blue dots, respectively.

### Local field potential simulation

LFP recordings were simulated by coupling the specific VC model variant with the neuron source models using a reciprocity-based solution [5]. In the coupled model system, each compartment (365) of each neuron model (221,601) was represented as an independent time-varying current source (80,884,365 total sources) at the appropriate spatial location of the VC model. The LFP recording at a DBS electrode contact was then calculated by summing the voltages imposed upon that contact from all of the transmembrane currents. Differential recordings for any bipolar pair of contacts could then be defined by subtracting the time series voltage signal recorded at one contact from the time series voltage signal recorded at the other contact. Identical time-varying current sources for each neuron model were used when evaluating each VC model variant.

### Results

The goal of this study was to quantify the effects of VC model complexity on simulated subthalamic LFPs recorded with DBS electrodes. We started our analyses with an idealized 2 mm radius volume of beta-synchronous neurons in the center of the STN volume. Contact 1 of the DBS electrode was positioned near the center of that synchronous volume, and in a typical location for therapeutic stimulation. We defined the most detailed model, using the MIDA12 anisotropic VC model, as the standard for comparison across the different models. Fig 2 displays example results from bipolar recording between contact 1 and 3 of the DBS electrode. Given the electrostatic simplifying assumptions employed in the finite element models (see Discussion), the different representations of the tissue medium in the VC models only impacted the amplitude of the simulated LFP signal. Therefore, our analyses concentrated on the peak-to-peak voltage recorded at the DBS electrode contacts. The LFP signal recorded when using MIDA12 anisotropic was the largest of the VC model variants, while MIDA2 was the smallest (Fig 2). For example, MIDA2 exhibited a 48.5% error in LFP signal amplitude relative to MIDA12 anisotropic for an idealized 2 mm radius volume of beta-synchronous neurons in the center of the STN volume.

The exact location of beta-synchronous volumes of neurons in the STN is a topic of scientific and clinical interest, but is not explicitly known in any given patient, and is likely to be variable from patient-to-patient. Therefore, we expanded our model analyses by moving the

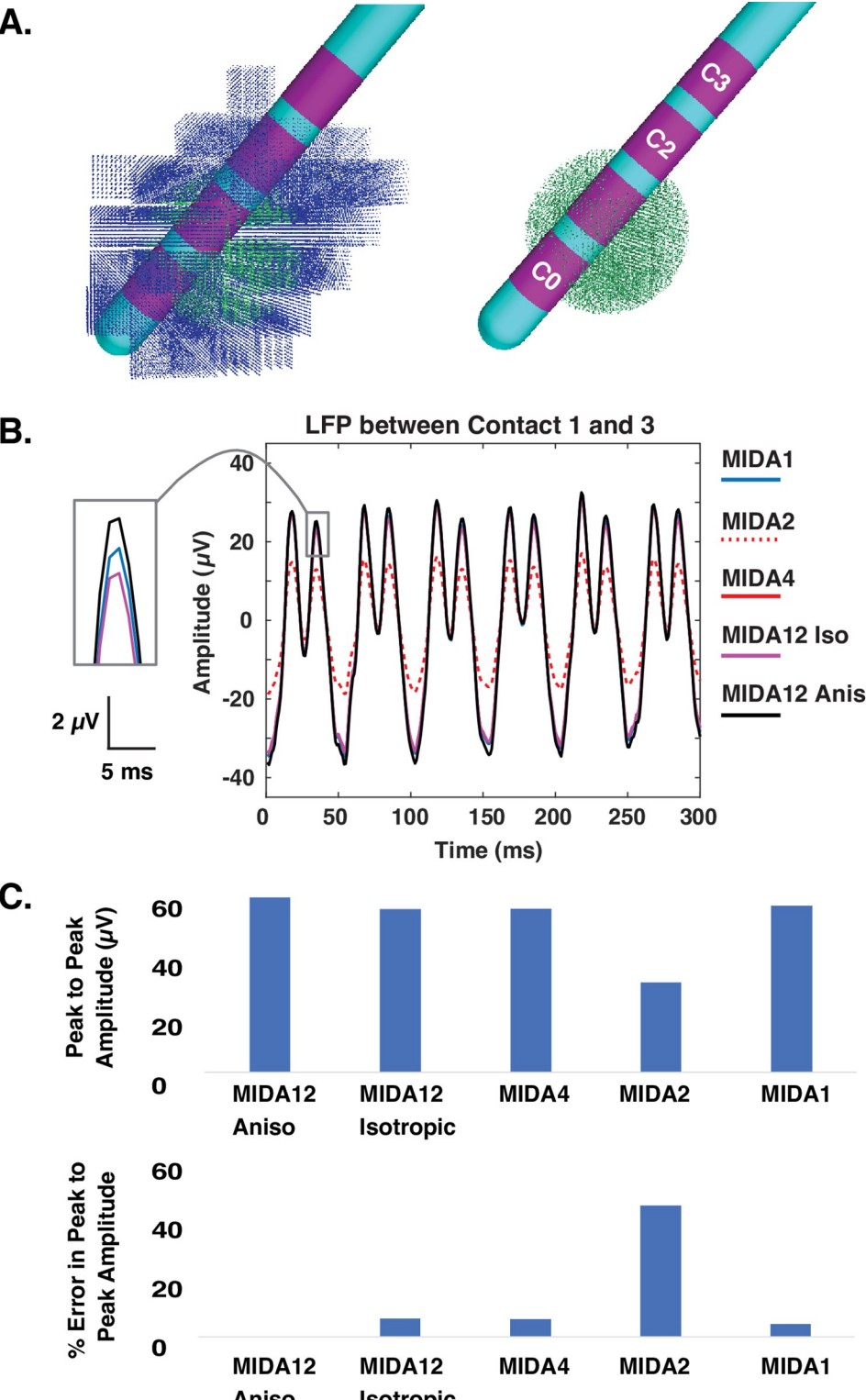

**Fig 2. LFP model.** A) Synchronous population of neurons (green dots) located in the center of the STN (with and without asynchronous neurons (blue dots)). B) Simulated LFP between contact 1 and 3 of the DBS electrode using various VC models. C) Peak-to-peak LFP amplitude (upper panel) and percent error in peak-to-peak amplitude (lower panel) obtained using different VC model variants compared to MIDA12 anisotropic.

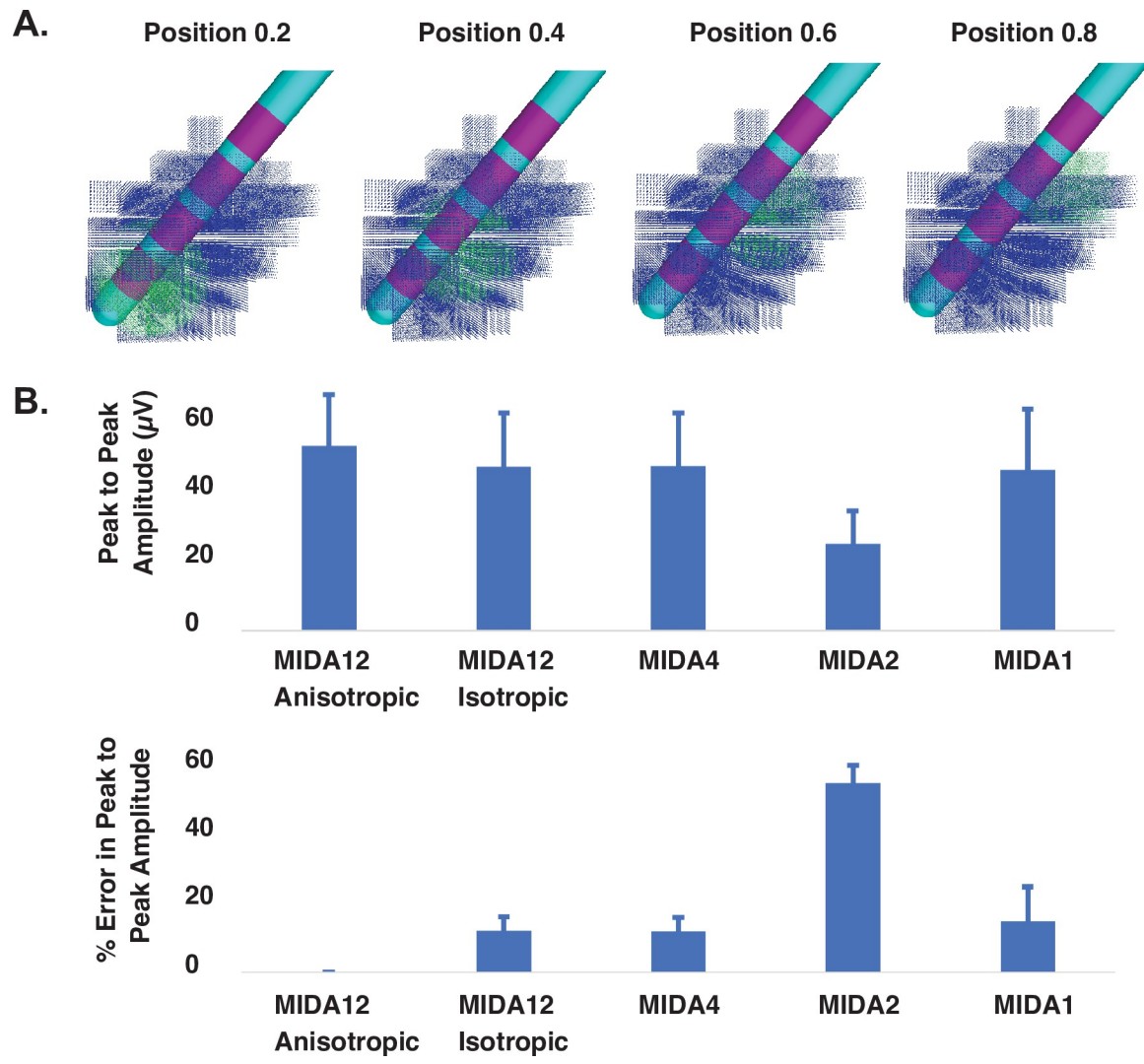

**Fig 3. LFP model.** A) Synchronous neuron population (green dots) at four different locations along the longitudinal axis of the STN. B) Upper panel shows mean peak-to-peak LFP amplitude (± SD) between contact 1 and 3 of the DBS electrode using different VC models. Lower panel shows mean percent error (± SD) in the peak-to-peak LFP amplitude, shown in the upper panel, with respect to MIDA12 anisotropic.

volume of beta-synchronous STN neurons to different locations within the STN volume (Fig 3). We computed the bipolar LFP between contacts 1 and 3 as the synchronous volume was moved along the longitudinal axis of the STN. The peak-to-peak LFP voltage, and percent error relative to the MIDA12 anisotropic, were calculated for each VC model variant at 4 different positions for the synchronous volume of neurons. Fig 3 shows the mean and standard deviations of the peak-to-peak voltage and percent error of each VC model variant. Averaged data show that the simulated LFP signal amplitude was largest with MIDA12 anisotropic. MIDA12 isotropic and MIDA4 exhibited the lowest relative errors (11.8% ± 4 and 11.7% ± 4), respectively.

Our results suggest that tissue anisotropy had a relatively small effect (MIDA12 anisotropic vs MIDA12 isotropic), while soft tissue heterogeneity had almost no effect (MIDA12 isotropic vs MIDA4), on the simulated LFP signal. However, the transition from MIDA4 to MIDA2,

which eliminated explicit representation of grey matter, white matter, and CSF regions in the model had a much larger effect on the simulated LFP. The reduced LFP signal amplitude observed with MIDA2 translated into an error of 53.2% ± 5.2. The transition from MIDA2 to MIDA1, which consolidated the entire head into a single isotropic volume increased the LFP signal amplitude into a range that was more consistent with the MIDA12 VC models.

Historically, simplified VC models with DBS electrodes have ignored the shape of the head and placed the electrode within a homogeneous medium in the shape of a box or cylinder. Therefore, we also tested if explicitly representing the head shape, as in MIDA 1, had any effect on the LFP simulation. A box shaped homogeneous VC model produced same results as MIDA1. This suggest that the general shape of the VC model boundaries do not influence the simulated LFP signal recorded with DBS electrodes.

## Discussion

Subthalamic LFP recordings have played an important role in advancing scientific understanding of the pathophysiology of Parkinson's disease, and are beginning to be used as electrophysiological biomarkers in adaptive DBS control systems [4]. The ever-growing importance of LFPs in DBS research has also prompted interest in developing a more detailed biophysical understanding of the recorded signals. Computational models provide opportunities to examine hypotheses on how synaptic activity and neural firing patterns work together to create an LFP signal [24]. However, the results of complex models that attempt to simulate detailed experiments are often sensitive to the specific computational methods employed when constructing the model system. Therefore, the goal of this study was to examine the role of the VC model on the simulation of beta-band (12–30 Hz) LFP activity recorded from subthalamic DBS electrodes.

Several previous studies have coupled multi-compartment STN neuron models with finite element based VC models to simulate LFP recordings acquired with DBS electrodes [5–7]. These studies assumed that the brain could be modeled as a homogenous and isotropic tissue medium (i.e. MIDA1 in this study). This general simplification is commonly applied in models of neural recording because the sources are in such close proximity to the recording electrodes that the nuances of the tissue medium are considered irrelevant [24]. However, incorporating tissue heterogeneity and white matter anisotropy into VC models can dramatically influence simulation results when quantifying the neural response to subthalamic DBS [8]. Therefore, we were motivated to better understand the implications of using typical simplifying assumptions for the tissue medium in subthalamic LFP modeling.

We relied on a highly detailed VC model to represent our standard for comparison (MIDA12 anisotropic). In general, reductions in the VC model complexity resulted in relatively limited errors (~15%) in the peak-to-peak LFP amplitude (i.e. MIDA12 anisotropic reduced to MIDA1). However, one of the simplified VC models (MIDA2) exhibited large errors (~53%). This drastic increase in error when the VC was simplified from four to two tissue types was because of the increase in the weighted-average tissue conductivity. In MIDA2, the weighted-average conductivity of grey matter, white matter, and CSF was 0.4 S/m. This value is almost twice the mean conductivity of grey and white matter, which are the two main tissue types surrounding the electrode contacts. We adjusted the conductivity of the interface layer surrounding the electrode to maintain the contact impedance at a consistent value of 1084 Ω across all the VC models (Table 2), but the MIDA2 model still suffered from large errors. This suggests that the conductivity of the tissue in close proximity of the DBS electrode is a major driver of the amplitude of the recorded signal. Our results are also consistent with previous findings that VC models with no distinction among CSF, grey matter, and white

matter performed worst among other models when attempting to simulate EEG recordings [Ramon et al., 2006]. Therefore, simply matching the electrical load of the volume conductor to a representative value (i.e. ~1KΩ for a cylindrical DBS electrode contact) does not necessarily guarantee accurate results.

Several different groups have investigated how tissue resistivity and anisotropy influence electric fields in the human head [9, 10, 25, 26]. These studies have found varying results that depend upon the focus of their analyses (i.e. EEG, MEG, TES, etc.). Our results suggest that tissue anisotropy and inhomogeneity are not major factors in subthalamic LFP simulations. Eliminating anisotropy did reduce the LFP signal amplitude by ~12% (i.e. MIDA12 anisotropic compared to MIDA12 isotropic) (Figs 2 and 3). This effect was dictated by the loss of the strong anisotropy in the internal capsule, just lateral to the STN, which acts to constrain decay of electric fields from the subthalamic region [27]. Subsequently eliminating the tissue heterogeneity had little additional effect on the simulated LFP (i.e. MIDA12 isotropic compared to MIDA1) (Figs 2 and 3). However, one reason we did not observe a significant impact of white matter anisotropy in our LFP simulations is that the neural sources and recording electrodes were both positioned within the STN volume. However, if the DBS lead was misplaced into the internal capsule, the errors become notably more pronounced (S1 Fig in S1 File).

LFP modeling studies often use a simplified VC model with a homogenous tissue conductivity of 0.3 S/m [28, 29]. However, our study was the first to directly compare LFP simulation results from homogenous VC models with results from a highly detailed VC model (Fig 3). Our analyses suggest that 0.215 S/m is a better average brain conductivity approximation for simplified VC models. In addition, simplified VC models like MIDA1 do appear to be reasonable options for DBS LFP modeling when the mean conductivity of the tissue medium is appropriately parameterized. However, if both stimulation and recording from DBS electrodes are intended to be simulated within a unified model, then a detailed VC model would be warranted, as tissue heterogeneity and anisotropy have a profound impact on the neural responses to stimulation [8].

An important limitation of this study was our use of electrostatic VC models. In reality, LFP signals are time varying, and the tissue medium can act as dielectric. We have previously demonstrated that the amplitude difference between DBS LFP simulations that explicitly account for the capacitance of the electrode-electrolyte interface layer, as well as the capacitance of the bulk brain tissue, is only ~3% when compared with electrostatic VC models [5]. These theoretical calculations are also consistent with experimental measurements demonstrating that the impedance of brain tissue is largely resistive and can be assumed to be frequency independent [30, 31]. Nonetheless, electrostatic LFP models have a known deficiency in their ability to create the 1/f scaling phenomenon observed in experimental LFP measurements [32]. In turn, it is possible to simulate 1/f behavior in LFP models by calculating the Fourier transform of the total membrane current for each current source in the model system (e.g. ~81M individual sources in our DBS LFP model), then calculating the Fourier component at each frequency, and then deriving the extracellular potential from the inverse Fourier transform. This process is associated with massive computational costs, but we have performed some simulations using this approach, and only found a modest effect of permittivity and heterogeneity on 1/f scaling (S2 Fig in S1 File). As such, 1/f behavior appears to be primarily dictated by ionic diffusion, which is also consistent with the conclusions from other recent studies [33, 34].

If computational costs are an important consideration for the LFP modeling analyses, then future efforts should focus on simplifying the representation of the neural sources. The neuron source models used in this study were an extreme example of detail, with explicit

representation of individual neurons at histologically defined densities surrounding the DBS electrode [6]. However, the millions of current source components in the human STN can likely be consolidated into several thousand simplified source models, each anatomically distributed within the STN volume, to generate LFP simulations that mimic the detailed model with reasonable fidelity. Such an approach could speed up the computations by orders of magnitude, which would be advantageous for inverse source localization modeling. Therefore, the next publication in this series will document our attempts to simplify the neural source models to facilitate high throughput LFP modeling of clinical recordings from STN DBS devices.

## Conclusion

We evaluated a range of different human head VC models for use in the simulation of subthalamic LFPs recorded with DBS electrodes. We compared LFPs obtained using each VC model variant with the most detailed model (MIDA12 anisotropic), and found that ignoring tissue anisotropy produced ~12% error in LFP amplitudes, while soft tissue heterogeneity had a negligible effect on the results. We also found that an isotropic VC model with a homogeneous conductivity of 0.215 S/m produced similar results as an isotropic VC model with 12 tissue types. In turn, the results of this study provide justification for the continued use of relatively simplified VC models in DBS LFP simulations.

## Supporting information

**S1 File. Contains supporting figures and supporting tables.**
(PDF)

## Acknowledgments

The authors thank Angela Noecker for her assistance with data visualization for this study.

## Author Contributions

**Conceptualization:** M. Sohail Noor, Cameron C. McIntyre.

**Data curation:** M. Sohail Noor.

**Formal analysis:** M. Sohail Noor.

**Funding acquisition:** Cameron C. McIntyre.

**Investigation:** M. Sohail Noor, Bryan Howell.

**Methodology:** M. Sohail Noor, Bryan Howell.

**Software:** Bryan Howell.

**Supervision:** Cameron C. McIntyre.

**Writing – original draft:** M. Sohail Noor.

**Writing – review & editing:** M. Sohail Noor, Bryan Howell, Cameron C. McIntyre.

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
