## [Decision Letter · Decision Letter 0]

11 Jul 2023

PONE-D-23-09459Role of the volume conductor on simulations of local field potential recordings from deep brain stimulation electrodesPLOS ONE

Dear Dr. McIntyre,

Thank you for submitting your manuscript to PLOS ONE. After careful consideration, we feel that it has merit but does not fully meet PLOS ONE’s publication criteria as it currently stands. Therefore, we invite you to submit a revised version of the manuscript that addresses the points raised during the review process.

We look forward to receiving your revised manuscript.

Kind regards,

Luca Aquili

Academic Editor

PLOS ONE

“I have read the journal's policy and the authors of this manuscript have the following competing interests: CCM is a paid consultant for Boston Scientific Neuromodulation, receives royalties from Hologram Consultants, Neuros Medical, Qr8 Health, and is a shareholder in the following companies: Hologram Consultants, Surgical Information Sciences, BrainDynamics, CereGate, Autonomic Technologies, Cardionomic, Enspire DBS.”

“This work was supported by a grant from the National Institutes of Health (R01 NS119520).  The funders had no role in study design, data collection and analysis, decision to publish, or preparation of the manuscript.”

“This work was supported by a grant from the National Institutes of Health (R01

NS119520). The funders had no role in study design, data collection and analysis,

decision to publish, or preparation of the manuscript.”

6. We note that Figure 1 in your submission contain copyrighted images. All PLOS content is published under the Creative Commons Attribution License (CC BY 4.0), which means that the manuscript, images, and Supporting Information files will be freely available online, and any third party is permitted to access, download, copy, distribute, and use these materials in any way, even commercially, with proper attribution. For more information, see our copyright guidelines: http://journals.plos.org/plosone/s/licenses-and-copyright.

Reviewers' comments:

Reviewer's Responses to Questions

**Comments to the Author**

1. Is the manuscript technically sound, and do the data support the conclusions?

Reviewer #1: Yes

Reviewer #2: Yes

Reviewer #3: Yes

2. Has the statistical analysis been performed appropriately and rigorously? 

Reviewer #1: Yes

Reviewer #2: N/A

Reviewer #3: Yes

3. Have the authors made all data underlying the findings in their manuscript fully available?

Reviewer #1: No

Reviewer #2: Yes

Reviewer #3: Yes

4. Is the manuscript presented in an intelligible fashion and written in standard English?

Reviewer #1: Yes

Reviewer #2: Yes

Reviewer #3: Yes

5. Review Comments to the Author

Reviewer #1: The data sharing method proposed by the authors (email address of the first author) appears equivalent to providing the data upon request and therefore inconsistent with PloS requirements. These requirements explicitly state that "provided upon request" is not an acceptable form of data sharing and asks that authors provide the data as part of the paper or make it available at URL/database location, failing which they need to explain why these options are not possible. The authors should therefore revise the data sharing section so as to conform with PloS requirements/recommendations.

Reviewer #2: The paper “Role of the Volume Conductor on Simulations of Local Field Potential

Recordings from Deep Brain Stimulation Electrodes” reports the results of a project aimed to study the effect of volume conductor definition, in terms of complexity, on LFP simulations. These results may be important considering the increasing interest in LFP analysis for new adaptive DBS strategies. The results suggest that, although simplified VC models introduce errors as compared to anisotropic complex models, they provide acceptable LFP simulations. The paper is well written and describes the experiment in details, with an in-depth discussion.

I have just a couple of points:

-The authors in the methods describe two different populations, one with synchronous input and one with asynchronous inputs. The asynchronous population was never mentioned again in the manuscript, likely because they do not produce a LFP. It is unclear to me why the authors mention this population in the methods.

-While the results on simulations of LFPs suggest that the increased VC anisotropy has little influence on the results, the authors underline that “VC models can dramatically influence simulation results when quantifying the neural response to subthalamic DBS”. Considering that the main expected application of LFP simulations is in the context of adaptive DBS, when LFP response to DBS is used to change simulation parameters, how this observation would impact the future application of the simulation technique. This could be addressed in the discussion, but also, if possible, with an additional experiment that includes the stimulation delivery.

Reviewer #3: The manuscript is well written and clear. The topic is of definite interest to the DBS field. The authors adequately responded to prior reviewer comments. I have only several comments:

Major:

The source of beta signal and recording location are basically the same, in the STN. It seems intuitive that in that case the tissue characteristics outside that volume would have minimal impact on the simulated LFPs which are so close to the source. It would be interesting to know what the results would be if the source was further away (and/or the source stayed the same in the STN and lead was placed further away, in the internal capsule for instance) and whether the conclusion about minimal impact of tissue heterogeneity still holds. This has relevance to clinical practice if leads are misplaced, or if beta signal generators are outside the STN. To me these results as presented are somewhat trivial, though I acknowledge they are worth establishing in the literature. I understand that in this specific context with recording location and signal source in the same location tissue homogeneity across the brain can be used, but I am left wondering, is there a situation where tissue anisotropy does matter for LFP recordings?

Minor:

The Methods are described in text and with numerous references. However I think it would be appropriate for reproducibility and clarity for the relevant modeling equations and parameters used to be included in the supplementary information.

MIDA2 produced the greatest error. It consolidated the CSF, grey and white matters together, but with a conductivity that was much higher than any individual component in MIDA 12 or MIDA4. This was mentioned in the Discussion, but it is not clear why such a large value was used?

6. PLOS authors have the option to publish the peer review history of their article (what does this mean?). If published, this will include your full peer review and any attached files.

Reviewer #1: No

Reviewer #2: No

Reviewer #3: No

---

## [Author Response · Author response to Decision Letter 0]

27 Sep 2023

PONE-D-23-09459

Role of the volume conductor on simulations of local field potential recordings from deep brain stimulation electrodes

Response to Reviewer Reports

Reviewer #1

The data sharing method proposed by the authors (email address of the first author) appears equivalent to providing the data upon request and therefore inconsistent with PLoS requirements. These requirements explicitly state that "provided upon request" is not an acceptable form of data sharing and asks that authors provide the data as part of the paper or make it available at URL/database location, failing which they need to explain why these options are not possible. The authors should therefore revise the data sharing section so as to conform with PLoS requirements/recommendations.

Response: We agree with the Reviewer. The simulation data and custom scripts used are now posted on GitHub - https://github.com/msohailnoor/LFPs-simulated-using-different-volume-conductor-models

Reviewer #2

The paper “Role of the Volume Conductor on Simulations of Local Field Potential Recordings from Deep Brain Stimulation Electrodes” reports the results of a project aimed to study the effect of volume conductor definition, in terms of complexity, on LFP simulations. These results may be important considering the increasing interest in LFP analysis for new adaptive DBS strategies. The results suggest that, although simplified VC models introduce errors as compared to anisotropic complex models, they provide acceptable LFP simulations. The paper is well written and describes the experiment in details, with an in-depth discussion. I have just a couple of points:

1) The authors in the methods describe two different populations, one with synchronous input and one with asynchronous inputs. The asynchronous population was never mentioned again in the manuscript, likely because they do not produce a LFP. It is unclear to me why the authors mention this population in the methods.

Response: The reviewer is correct that LFP signals primarily result from the synchronous neurons. However, to enhance the realism of our model, we included both synchronous and asynchronous population in the model. The currents generated by asynchronous neurons contribute to the noise in the LFP signal. This point has been added to the Methods (p. 6)

2) While the results on simulations of LFPs suggest that the increased VC anisotropy has little influence on the results, the authors underline that “VC models can dramatically influence simulation results when quantifying the neural response to subthalamic DBS”. Considering that the main expected application of LFP simulations is in the context of adaptive DBS, when LFP response to DBS is used to change simulation parameters, how this observation would impact the future application of the simulation technique. This could be addressed in the discussion, but also, if possible, with an additional experiment that includes the stimulation delivery.

Response: We agree with the Reviewer’s point. The technical challenges of accurately simulating the LFP during DBS in a complex model system like this are daunting to say the least, and beyond the scope of this paper. However, we have added the basic point that future efforts to study simultaneous stimulation and recording should focus on using the MIDA12 anisotropic VC (or similarly detailed VC model) (p. 9).

Reviewer #3

The manuscript is well written and clear. The topic is of definite interest to the DBS field. The authors adequately responded to prior reviewer comments. I have only several comments:

Major:

The source of beta signal and recording location are basically the same, in the STN. It seems intuitive that in that case the tissue characteristics outside that volume would have minimal impact on the simulated LFPs which are so close to the source. It would be interesting to know what the results would be if the source was further away (and/or the source stayed the same in the STN and lead was placed further away, in the internal capsule for instance) and whether the conclusion about minimal impact of tissue heterogeneity still holds. This has relevance to clinical practice if leads are misplaced, or if beta signal generators are outside the STN. To me these results as presented are somewhat trivial, though I acknowledge they are worth establishing in the literature. I understand that in this specific context with recording location and signal source in the same location tissue homogeneity across the brain can be used, but I am left wondering, is there a situation where tissue anisotropy does matter for LFP recordings?

Response: We agree with the Reviewer’s point. Therefore, we performed a series of additional simulations with the DBS electrode “misplaced” into the internal capsule (4mm lateral to the default position in the STN volume). As expected, the errors in peak-to-peak amplitude of the simplified models were greater with the DBS lead in the IC. These results are now included in the Supplemental Material (Figure S1) and this point is stated in the Discussion (p. 9).

Minor:

The Methods are described in text and with numerous references. However I think it would be appropriate for reproducibility and clarity for the relevant modeling equations and parameters used to be included in the supplementary information.

Response: The Supplementary Material has been updated with the neuronal densities for each sector of the STN (Table S1), conductance of each ion channel (Table S2), and the Python functions used for generating jitter in synaptic input timing for synchronous and asynchronous neurons.

MIDA2 produced the greatest error. It consolidated the CSF, grey and white matters together, but with a conductivity that was much higher than any individual component in MIDA12 or MIDA4. This was mentioned in the Discussion, but it is not clear why such a large value was used?

Response: The lumped conductivities were determined as the weighted-average of the conductivity of the combined tissue types. For instance, in MIDA2, the conductivities of CSF, grey matter, and white matter were 1.5, 0.23, and 0.14 S/m, respectively (column 3, Table 1). Likewise, the percentages of CSF, grey matter, and white matter were 5.3%, 14.85%, and 12.83%, respectively (column 2). The weighted-average conductivity of the lumped tissue, was calculated as:

(5.3 * 1.5 + 14.85 * 0.23 + 12.83 * 0.14) / (5.3 + 14.85 + 12.83) = 0.4 S/m.

Please note that the resulting value of 0.4 S/m (column 5) is lower than the individual conductivity of CSF (1.5 S/m), it was still skewed by the CSF, resulting in a large lumped conductivity value.

---

## [Decision Letter · Decision Letter 1]

3 Nov 2023

Role of the volume conductor on simulations of local field potential recordings from deep brain stimulation electrodes

PONE-D-23-09459R1

Dear Dr. McIntyre,

We’re pleased to inform you that your manuscript has been judged scientifically suitable for publication and will be formally accepted for publication once it meets all outstanding technical requirements.

Kind regards,

Luca Aquili

Academic Editor

PLOS ONE

Additional Editor Comments (optional):

Reviewers' comments:

Reviewer's Responses to Questions

**Comments to the Author**

1. If the authors have adequately addressed your comments raised in a previous round of review and you feel that this manuscript is now acceptable for publication, you may indicate that here to bypass the “Comments to the Author” section, enter your conflict of interest statement in the “Confidential to Editor” section, and submit your "Accept" recommendation.

Reviewer #3: All comments have been addressed

2. Is the manuscript technically sound, and do the data support the conclusions?

Reviewer #3: Yes

3. Has the statistical analysis been performed appropriately and rigorously? 

Reviewer #3: Yes

4. Have the authors made all data underlying the findings in their manuscript fully available?

Reviewer #3: Yes

5. Is the manuscript presented in an intelligible fashion and written in standard English?

Reviewer #3: Yes

6. Review Comments to the Author

Reviewer #3: Comments were adequately addressed. I think the added experiments would probably be more appropriate in the results section, and then also discussed in the discussion section, but this is only a suggestion.

7. PLOS authors have the option to publish the peer review history of their article (what does this mean?). If published, this will include your full peer review and any attached files.

Reviewer #3: No

---

## [Editor Report · Acceptance letter]

14 Nov 2023

PONE-D-23-09459R1 

Role of the volume conductor on simulations of local field potential recordings from deep brain stimulation electrodes 

Dear Dr. McIntyre:

I'm pleased to inform you that your manuscript has been deemed suitable for publication in PLOS ONE. Congratulations! Your manuscript is now with our production department. 

Kind regards, 

on behalf of

Dr. Luca Aquili 

Academic Editor

PLOS ONE